# Accurate timekeeping is controlled by a cycling activator in *Arabidopsis*

**Polly Yingshan Hsu, Upendra K Devisetty, Stacey L Harmer\***

Department of Plant Biology, University of California, Davis, Davis, United States

**Abstract** Transcriptional feedback loops are key to circadian clock function in many organisms. Current models of the *Arabidopsis* circadian network consist of several coupled feedback loops composed almost exclusively of transcriptional repressors. Indeed, a central regulatory mechanism is the repression of evening-phased clock genes via the binding of morning-phased Myb-like repressors to evening element (EE) promoter motifs. We now demonstrate that a related Myb-like protein, REVEILLE8 (RVE8), is a direct transcriptional activator of EE-containing clock and output genes. Loss of RVE8 and its close homologs causes a delay and reduction in levels of evening-phased clock gene transcripts and significant lengthening of clock pace. Our data suggest a substantially revised model of the circadian oscillator, with a clock-regulated activator essential both for clock progression and control of clock outputs. Further, our work suggests that the plant clock consists of a highly interconnected, complex regulatory network rather than of coupled morning and evening feedback loops.

## Introduction

Circadian clocks are widespread in nature, presumably because they help diverse organisms prepare for predictable day/night cycles. Although specific components are not widely conserved, eukaryotic clocks are composed of interlocking negative transcriptional feedback loops (*Harmer, 2009*). In Arabidopsis, the first-identified clock genes function in a double negative feedback loop, with two morning-phased Myb-like transcription factors, CIRCADIAN CLOCK ASSOCIATED 1 (CCA1) and LATE ELONGATED HYPOCOTYL (LHY), repressing expression of an evening-phased pseudo-response regulator, TIMING OF CAB EXPRESSION 1 (TOC1 or PRR1), which in turn represses expression of *CCA1* and *LHY* (*Schaffer et al., 1998*; *Wang and Tobin, 1998*; *Strayer et al., 2000*; *Alabadi et al., 2001*; *Gendron et al., 2012*; *Huang et al., 2012*; *Pokhilko et al., 2012*). CCA1 and LHY also promote the expression of *PRR7* and *9*, two day-phased genes, and are in turn repressed by these PRRs and their homolog PRR5, forming another negative feedback circuit (*Farre et al., 2005*; *Nakamichi et al., 2010*). Finally, TOC1, GIGANTEA (GI), and the evening complex components including LUX ARRHYTHMO (LUX), EARLY FLOWERING 3 (ELF3) and 4 (ELF4), act in double negative feedback loops with CCA1, LHY, and PRR7 and 9 (*Fowler et al., 1999*; *Park et al., 1999*; *Dixon et al., 2011*; *Helfer et al., 2011*; *Nusinow et al., 2011*; *Huang et al., 2012*; *Pokhilko et al., 2012*). Thus most characterized clock components repress expression of other clock components.

A *cis*-regulatory element named the evening element (EE) [(A)AAATATCT] has been found to be central to circadian clock function in plants. Most evening-phased central clock genes (including *TOC1*, *PRR5*, *GI*, *LUX* and *ELF4*) contain the EE in their promoter regions (*Covington et al., 2008*; *Harmer, 2009*) and the two morning-phased components, CCA1 and LHY, bind directly to the EE to repress evening-phased clock gene expression (*Alabadi et al., 2001*). The EE was first identified by its over-representation in the promoters of evening-phased genes (*Harmer et al., 2000*) and is sufficient to confer evening-phased expression on a reporter gene (*Harmer and Kay, 2005*). In addition to these two morning-phased transcriptional repressors that act via the EE, two pieces of evidence suggest that there is also a transcriptional activator(s) present in the afternoon that regulates the EE. First,

**\*For correspondence:**
slharmer@ucdavis.edu

**Competing interests:** The authors declare that no competing interests exist.

**Reviewing editor**: Joanne Chory, Salk Institute, United States

**eLife digest** We live in a world with a 24-hr cycle in which day follows night follows day with complete predictability. Life on earth has evolved to take advantage of this predictability by using circadian clocks to prepare for the coming of night (or day), and plants are no exception. Even in constant darkness, characteristics such as leaf movements show a constant cycle of around 24 hr.

Most circadian clocks rely on negative feedback loops involving various genes and proteins to keep track of time. In one of these feedback loops, certain genes—called morning-phased genes—are expressed as proteins during the day, and these proteins prevent other genes—called evening-phased genes—from producing proteins. As night approaches, however, a second feedback loop acts to stop the morning-phased genes being expressed, thus allowing the evening-phased genes to produce proteins. And as day approaches, expression of these genes is stopped and the whole cycle starts again.

Many of the genes and proteins involved in the circadian system of *Arabidopsis thaliana*, a small flowering plant that is widely used as a model organism, have been identified, and its circadian clock was thought to rely almost entirely on proteins called repressors that block the transcription of genes. Now, Hsu et al. have shown that the Arabidopsis clock also involves proteins that increase the expression of certain genes at specific times of the day.

Hsu et al. focused on the promoter regions of evening-phased genes: these regions are stretches of DNA that proteins called transcription factors bind to and either encourage the expression of a gene (if the protein is a transcriptional activator) or block its expression (as a transcriptional repressor). In particular, they focused on a protein called RVE8 that is most strongly expressed in the afternoon and, based on previous research, is thought to activate the transcription of genes. Using genetically modified plants in which the gene for RVE8 can be turned on and off, they found that this protein led to increases in the expression of some genes, and reductions in the expression of others.

Further analysis showed that RVE8 was able to activate the expression of evening-phased genes directly, without requiring that new proteins be made first. By contrast, morning-expressed genes were likely to be suppressed by RVE8 via an indirect mechanism that involved other proteins that had previously been activated by RVE8. The expression of RVE8 itself is regulated by other clock genes and also by an undefined post-transcriptional process. Therefore rather than consisting of a morning feedback loop coupled to an evening feedback loop, with both loops being based on repressors, the plant clock is instead better viewed as a highly connected network of activators and repressors. Further research is clearly necessary to understand this unexpected complexity in the circadian clock of Arabidopsis.

if only repressors bind to the EE, loss of protein binding to the EE should result in constitutively high expression of EE-regulated genes; however, mutation of the EE causes decreased expression of an EE-regulated reporter gene (*Harmer and Kay, 2005*). Second, an afternoon/evening-phased activity that specifically binds the EE is present in plant extracts and persists in *cca1 lhy* mutants, consistent with the existence of a clock-regulated, afternoon-phased activator of the EE (*Harmer and Kay, 2005*). A clock-regulated activator of the EE might help to explain why evening-phased clock genes are expressed with a circadian rhythm in *cca1 lhy* plants rather than being arrhythmic (*Mizoguchi et al., 2002*).

A candidate activator of the EE is REVEILLE 8/ LHY-CCA1-LIKE 5 (RVE8/LCL5) (*Farinas and Mas, 2011*; *Rawat et al., 2011*). RVE8 has been shown to bind to the EE in vitro and in planta, and its protein levels display a circadian rhythm that peaks in the afternoon (*Gong et al., 2008*; *Rawat et al., 2011*). Furthermore, *rve8* loss of function mutations cause a long circadian period (*Farinas and Mas, 2011*; *Rawat et al., 2011*) which is opposite to the phenotypes of *cca1* or *lhy* loss of function mutants (*Green and Tobin, 1999*; *Mizoguchi et al., 2002*). However, despite its ability to bind to the EE in the *TOC1* and *PRR5* promoters in planta, loss of RVE8 function does not significantly affect the transcript levels of these evening genes (*Farinas and Mas, 2011*; *Rawat et al., 2011*; *Hsu and Harmer, 2012*), perhaps due to genetic redundancy or complex feedback regulation within the clock system. Here, we used an inducible RVE8 line and genome-wide expression profiling to identify hundreds of

clock-regulated genes controlled by RVE8. Experiments with an inhibitor of translation revealed that most evening-phased clock genes are directly induced by RVE8. Consistent with RVE8 acting via the EE regulatory motif, we found that genes induced by RVE8 are enriched for the EE in their promoter regions. Furthermore, plants mutant for *RVE8* and its two closest homologs, *RVE4* and *RVE6*, have lost the afternoon-phased EE-binding activity. Finally, *rve4 rve6 rve8* triple mutants display an extremely long circadian period, with delayed and reduced expression of evening-phased clock genes. Together, these data suggest a considerably revised model of the plant clock, with an indispensable role for activators of transcription within the circadian regulatory network. Our work shows that rather than consisting of discrete, interlocked feedback loops, the plant circadian oscillator is more accurately described as a highly interconnected complex network.

## Results

### RVE8 activity is stronger in the afternoon

To identify RVE8 target genes, we generated a line with rapidly inducible RVE8 activity. A translational fusion between RVE8 and the glucocorticoid receptor (GR), driven by the native *RVE8* promoter, was introduced into *rve8-1* plants. GR fusion proteins are held in the cytoplasm unless the synthetic ligand for GR, dexamethasone (DEX), is applied, which allows the chimeric factor to move into the nucleus (*Picard et al., 1988*). Both *rve8-1* and *rve8-1 RVE8::RVE8:GR* plants have a long-period phenotype that is only rescued by DEX treatment of the *rve8-1 RVE8::RVE8:GR* line (*Figure 1A,B*), demonstrating that the RVE8:GR fusion protein retains RVE8 function and acts in a drug-inducible manner.

We next examined the ability of DEX-inducible RVE8-GR to activate expression of a known RVE8 target, the evening-phased clock gene *PRR5* (*Rawat et al., 2011*). Since RVE8 protein levels are

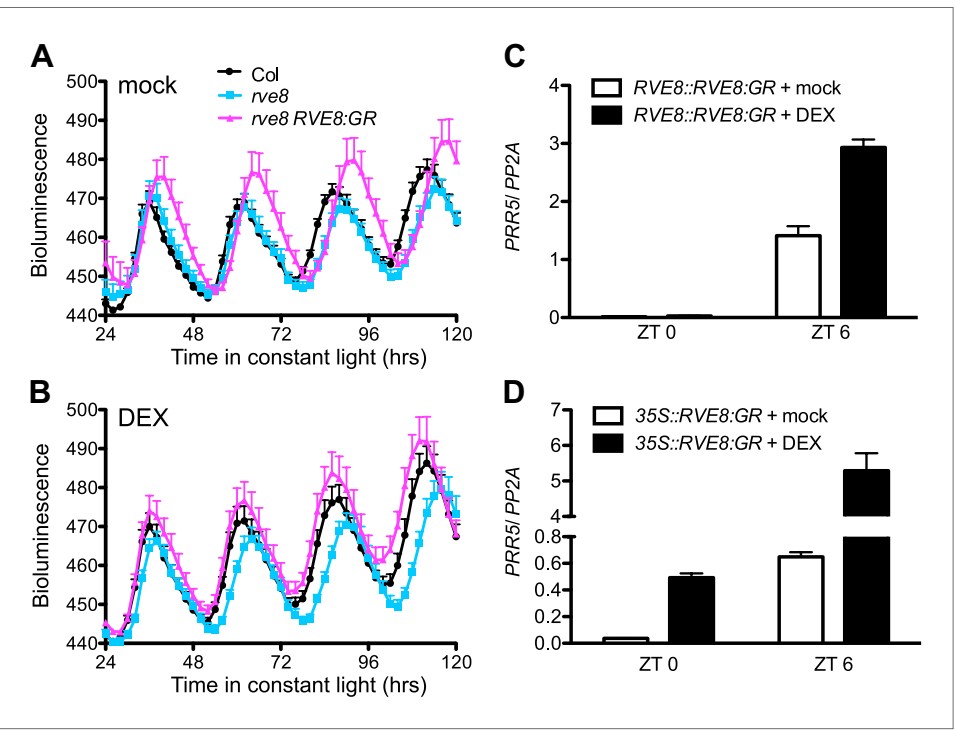

**Figure 1**. Activation of *PRR5* by RVE8 induction is stronger in the afternoon. (**A**) and (**B**) Luciferase activity in mock (**A**) and DEX-treated (**B**) Col, *rve8-1* and *rve8-1 RVE8::RVE8:GR* plants transgenic for the *CCR2::LUC* reporter. Plants were entrained in 12:12 light/dark (LD) cycles for 6 days and then sprayed with 30 µM DEX or 0.05% ethanol (mock treatment) plus luciferin before release to constant red light (30 µEi) for imaging of bioluminescence. Mean + SEM from 17 to 25 plants are represented. (**C**) and (**D**) Transcript levels of *PRR5* in response to induction of RVE8 activity in *rve8-1 RVE8::RVE8:GR* (**C**) or *rve8-1 35S::RVE8:GR* (**D**) at different time of day. 30 µM DEX or 0.05% ethanol (mock) was applied at the times indicated and the plants were harvested 2 hr later. Expression levels were quantified by qRT-PCR and normalized to *PP2A*. Mean ± SEM from three biological replicates are represented.

circadian-regulated, with peak protein abundance in the subjective afternoon (*Rawat et al., 2011*), we tested the ability of RVE8 to activate *PRR5* after DEX induction in the morning or afternoon. Induction of *PRR5* by RVE8 is much stronger when RVE8 activity is induced in the afternoon (Zeitgeber Time 6 [ZT6], or 6 hr after lights on) than when RVE8 is induced in the morning (ZT0) (*Figure 1C*). Similarly, although induction of constitutively expressed *RVE8* (*35S::RVE8:GR*) in the morning (ZT0) is sufficient to induce *PRR5*, this induction is much stronger when the DEX treatment is given in the afternoon (ZT6) (*Figure 1D*). These data indicate the ability of RVE8 to induce target genes is gated, with maximum activity in the afternoon.

## RVE8 preferentially regulates clock-controlled genes, inducing evening genes and repressing morning genes

To globally identify RVE8 target genes, we induced RVE8 activity near the time of normal peak RVE8 protein accumulation (*Figure 2A*) and used RNA-seq analysis to characterize the transcriptome in response to RVE8 induction (experimental design, *Figure 2B*; analysis summary, *Supplementary file 1A,B*). Verification of RNA-seq results using qRT-PCR showed excellent correlation between the two techniques, suggesting our RNA-seq results are reliable (*Figure 2—figure supplement 1*). Comparing mock- and DEX-treated *RVE8:GR* and *rve8-1* plants, we found that 583 genes are specifically up- and 850 are down-regulated in response to RVE8 induction (*Figure 2C,D* and *Supplementary file 1C–F*). Interestingly, a significantly higher proportion of both the up- and down-regulated RVE8 targets are clock-controlled (*Figure 2E,F*, 64% and 62%, respectively) than the one-third of the transcriptome expected by chance (*Covington et al., 2008*; *Hsu and Harmer, 2012*). RVE8 thus preferentially regulates clock-controlled genes (CCGs).

CCGs regulated by RVE8 are enriched for two complementary circadian phases, with the RVE8-induced genes enriched for an evening (*Figure 2G*) and the RVE8-repressed genes enriched for a morning phase (*Figure 2H*). Many evening-phased oscillator genes are induced by RVE8, including *PRR5*, *TOC1*, *PRR3*, *GI*, *LUX*, and *ELF4* (*Supplementary file 1G*). In contrast, morning-phased oscillator genes including *CCA1*, *LHY*, *RVE8* itself, and a day-phased central clock gene, *PRR9*, are found to be repressed by RVE8 (*Supplementary file 1G*). Activation of evening-phased and repression of morning-phased central clock genes suggests that RVE8 acts as a key regulator within the central system.

## EE promoter motifs are enriched among RVE8-induced target genes

To identify possible in vivo RVE8 binding sites, we identified promoter motifs found more frequently than expected by chance among the CCGs up- or down-regulated in response to RVE8 induction. EE and EE-like sequences are significantly overrepresented in the RVE8-induced CCGs, both when compared to their frequency in all genes in the genome (*Supplementary file 2A*) and in all evening-phased CCGs (*Table 1A*). This indicates that RVE8 preferentially regulates evening-phased genes containing an EE or EE-like promoter sequence. Since RVE8 directly binds to the EE in vitro and in vivo (*Rawat et al., 2011*), this suggests that RVE8 may directly activate many evening genes via binding to the EE in their promoters.

Among CCGs repressed by RVE8, we found motifs related to the G-box and morning element (ME) to be overrepresented when compared to all genes in the genome (*Supplementary file 2B*). Since most RVE8-repressed genes are also morning-phased CCGs (*Figure 2H*), we compared the frequency of these motifs between RVE8-repressed and all morning-phased CCGs. Unlike our results for the EE, the G-box and ME motifs are found at a similar rate in RVE8-repressed and in phase-matched CCGs (*Table 1B*). The similar frequency of these two motifs in these two groups indicates that RVE8 activity is not preferentially correlated with the morning-phased related *cis*-regulatory elements. The preferential correlation of RVE8 activity with the EE, but not with the morning-associated motifs, suggests that RVE8 may directly activate evening-phased clock genes that then go on to repress morning-phased CCGs.

## RVE8 directly activates evening genes but represses morning genes indirectly

To investigate whether RVE8 regulates morning and evening clock genes directly or indirectly, we induced RVE8 activity in the presence of cycloheximide (CHX), a protein synthesis inhibitor, and then examined transcript levels of genes identified as RVE8-regulated in our RNA-seq experiment. Genes

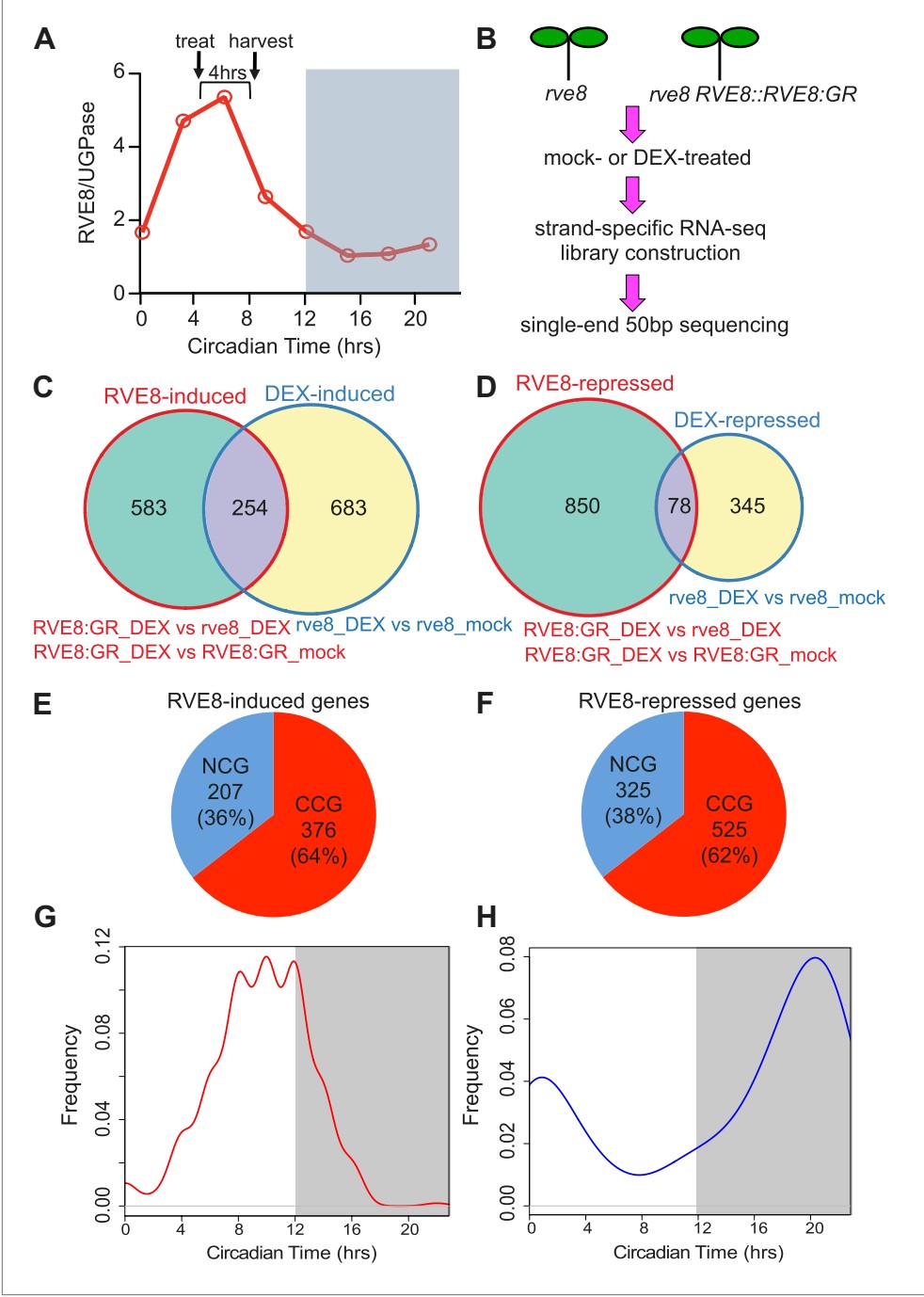

**Figure 2**. Identification of RVE8 targets by RNA-seq. RNA-seq experimental design and data analysis. (**A**) Relative timing of RVE8 induction and RVE8 protein abundance during a day. Adapted from **Rawat et al. (2011)**. (**B**) Scheme of experimental design. (**C**) and (**D**) Weighted Venn diagrams of genes significantly responsive to RVE8 induction and/or DEX treatment. Genes up-regulated (**C**) or down-regulated (**D**) by RVE8 and/or DEX. Differentially expressed genes were identified using edgeR (**Robinson et al., 2010**) with an adjusted p-value <0.01 as the cutoff. Genes significantly different between '*RVE8:GR* + DEX' and '*RVE8:GR* + mock' or between '*RVE8:GR* + DEX' and '*rve8* + DEX' are grouped into the RVE8-induced (**C**) or RVE8-repressed sets (**D**) shown in red circles. Genes significantly different between '*rve8* + DEX' and '*rve8* + mock' are grouped into the 'DEX-induced' (**C**) or 'DEX-repressed' (**D**) sets shown in blue circles. The genes uniquely induced or repressed by RVE8 (the 583 and 850 genes shown in green areas in (**C**) and (**D**), respectively) were defined as RVE8-regulated and used for further analysis. (**E**) and (**F**) The relative proportion of clock-controlled genes (CCGs) and non-clock-controlled genes
*Figure 2. Continued on next page*

*Figure 2. Continued*

(NCGs) among RVE8 targets. RVE8-induced genes (**E**); RVE8-repressed genes (**F**). (**G**) and (**H**) Circadian phase distributions of RVE8-regulated CCGs. CCGs up-regulated by RVE8 (**G**); CCGs down-regulated by RVE8 (**H**). White box: subjective day; grey box: subjective night. X-axis, 0: subjective dawn, 12: subjective dusk. Phase estimates are from previously published data (*Hsu and Harmer, 2012*). See also *Supplementary file 1*.
The following figure supplements are available for figure 2:

**Figure supplement 1**. Expression levels as determined by RNA-seq and qRT-PCR are highly correlated.

regulated by RVE8 both in the presence or absence of CHX would be considered direct targets while those only regulated by RVE8 in the absence of CHX would be considered indirect targets. CHX treatment increased the accumulation of transcripts regulated by the nonsense mediated mRNA decay (NMD) pathway (*Carter et al., 1995*; *Arciga-Reyes et al., 2006*; *Kurihara et al., 2009*), suggesting that CHX treatment reduced or blocked translation as expected (*Figure 3—figure supplement 1A–C*). Consistent with a role for RVE8 in activation of evening genes via direct binding to the EE, all of the EE-containing, evening-phased central clock and output genes examined are robustly induced by RVE8 even in the presence of CHX (*Figure 3A–F*). In contrast, the RVE8-mediated repression of expression of all tested morning genes is reduced or abolished in the presence of CHX (*Figure 3—figure supplement 1D–G*), suggesting that RVE8 represses these genes indirectly. In the case of *PRR9*, induction of RVE8 in the presence of CHX actually causes increased expression levels rather than the decrease seen in the absence of CHX (*Figure 3—figure supplement 1E*). RVE8-mediated activation of *PRR9* is likely masked in the absence of CHX by the concomitant induction of strong repressors of *PRR9* expression such as *TOC1* and *LUX* (*Helfer et al., 2011*; *Gendron et al., 2012*; *Huang et al., 2012*) (*Figure 3C,E*) and is only revealed when the translation of these repressors is blocked. In summary, for all of the genes examined, we found that RVE8 directly activates evening-phased genes but indirectly represses the morning-phased genes.

**Table 1.** Enrichment of EE, G-box-like and ME-like motifs in CCGs regulated by RVE8 compared to their occurrence in all CCGs previously defined as either evening-phased or morning-phased (*Hsu and Harmer, 2012*)

**(A) Evening-phased genes (CT 8 to CT 14)**

| | | CCGs (2709 genes) | | RVE8-induced CCGs (278 genes) | | |
|---|---|---|---|---|---|---|
| Motif | Sequence | Genes with the motif | Coverage (%) | Genes with the motif | Coverage (%) | p |
| Short EE | AAATATCT | 794 | 29.3 | 152 | 54.5 | $<2.2 \times 10^{-16}$*** |
| Long EE | AAAATATCT | 444 | 16.4 | 104 | 37.5 | $2.06 \times 10^{-15}$*** |
| EE-like | AATATCT | 1360 | 50.2 | 190 | 68.2 | $7.39 \times 10^{-09}$*** |

**(B) Morning-phased genes (CT 20 to CT 2)**

| | | CCGs (1572 genes) | | RVE8-repressed CCGs (328 genes) | | |
|---|---|---|---|---|---|---|
| Motif | Sequence | Genes with the motif | Coverage (%) | Genes with the motif | Coverage (%) | p |
| G-box-like | BACGTRD | 1187 | 75.5 | 266 | 81.0 | 0.0317* |
| ME-like | CCACA | 1429 | 90.9 | 308 | 93.9 | 0.08297 |

To determine whether the over-represented motifs found in RVE8 targets (*Supplementary file 2*) are enriched when compared to the morning-phased and evening-phased CCG groups, the number of genes containing the motif in each phase group was compared to that in the up- or down-regulated RVE8 targets. Fisher's exact test was performed to determine if the ratios in both groups are significantly different (*p<0.05; **p<0.01; ***p<0.001).

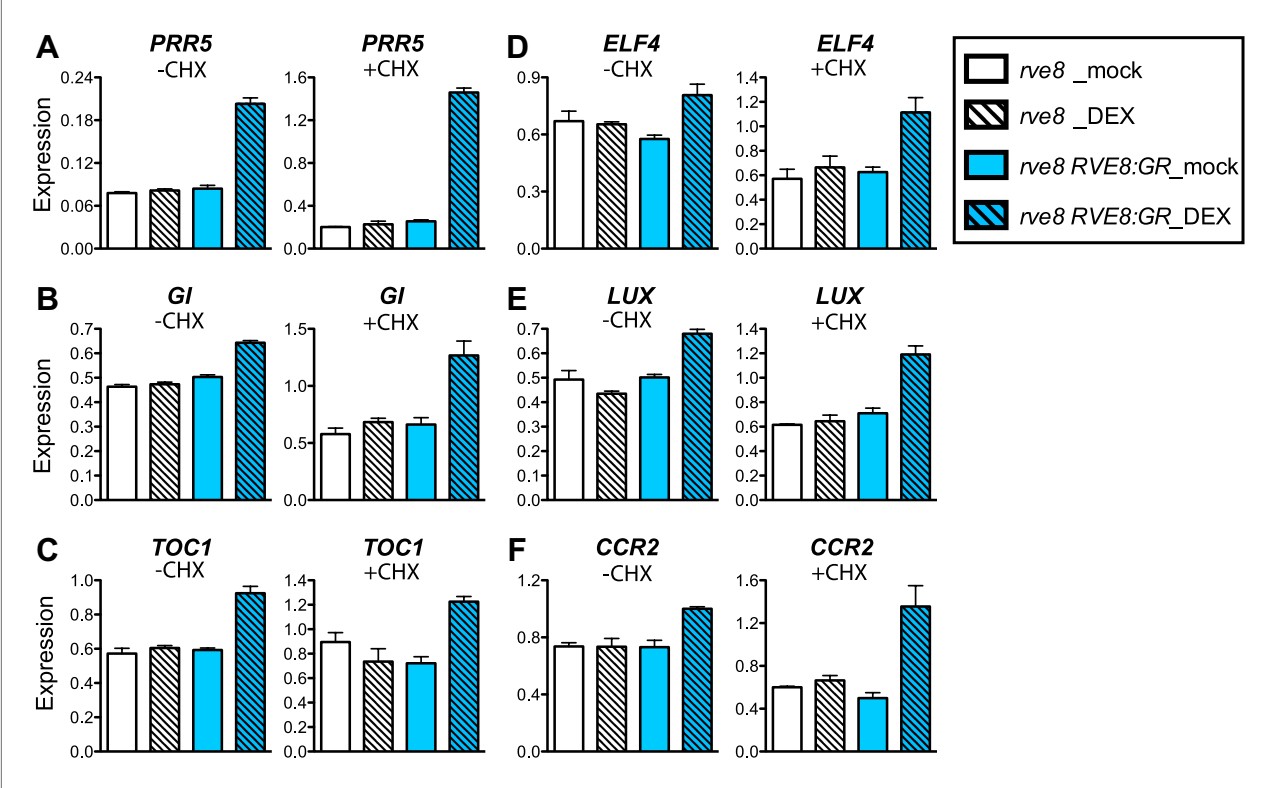

**Figure 3**. RVE8 activates evening genes directly. (**A**)–(**F**) Transcript levels of evening genes in response to RVE8 induction in the absence or presence of cycloheximide (CHX). 7-day-old *rve8-1* and *rve8-1 RVE8::RVE8:GR* plants were grown in light:dark (LD) cycles and mock- or DEX-treated in the absence or presence of CHX at ZT4 (4 hr after dawn) and harvested at ZT8 (8 hr after dawn). (**A**–**E**) Evening-phased clock genes. (**F**) Evening-phased clock output gene. Transcript levels were determined by qRT-PCR and then normalized to *PP2A*. Mean ± SEM from three biological replicates are represented.

The following figure supplements are available for figure 3:

**Figure supplement 1**. Morning-phased genes are indirectly repressed in response to RVE8 induction.

Since our data suggest that RVE8 is primarily (perhaps exclusively) an activator of transcription, we examined the physiological functions of all RVE8-induced genes in order to identify clock output pathways that may be directly influenced by RVE8. Functional classifications in which RVE8-induced genes are statistically overrepresented include regulation of the central oscillator (*Supplementary file 1H*), as expected given the clock phenotype of *rve8* plants (*Farinas and Mas, 2011*; *Rawat et al., 2011*; *Hsu and Harmer, 2012*). In addition, genes acting in pathways related to responses to the environments (including external stimulus, defense, temperature and stress), hormone regulation and metabolic processes are also enriched (*Supplementary file 1H*). Together, these data suggest that RVE8 shapes the evening-phased expression of hundreds of genes, directly influencing a large number of circadian output pathways.

## RVE8-family proteins act through the EE

Comparison of the phases of expression of RVE8-induced CCGs that have EE promoter motifs to the phases of all CCGs with EE sequences showed that the RVE8-regulated genes have a much narrower range of phases (*Figure 4A*). Almost all RVE8-induced EE-containing genes have peak expression in the subjective evening. Interestingly, the mean peak phase for RVE8-regulated EE-containing CCGs is significantly earlier than that of all EE-containing CCGs, indicating that RVE8 regulates a subset of evening genes that have slightly earlier phase than average EE-containing evening genes (*Figure 4A*). These data are consistent with the afternoon-phased RVE8 binding to the EE to induce expression of a subset of evening-phased genes.

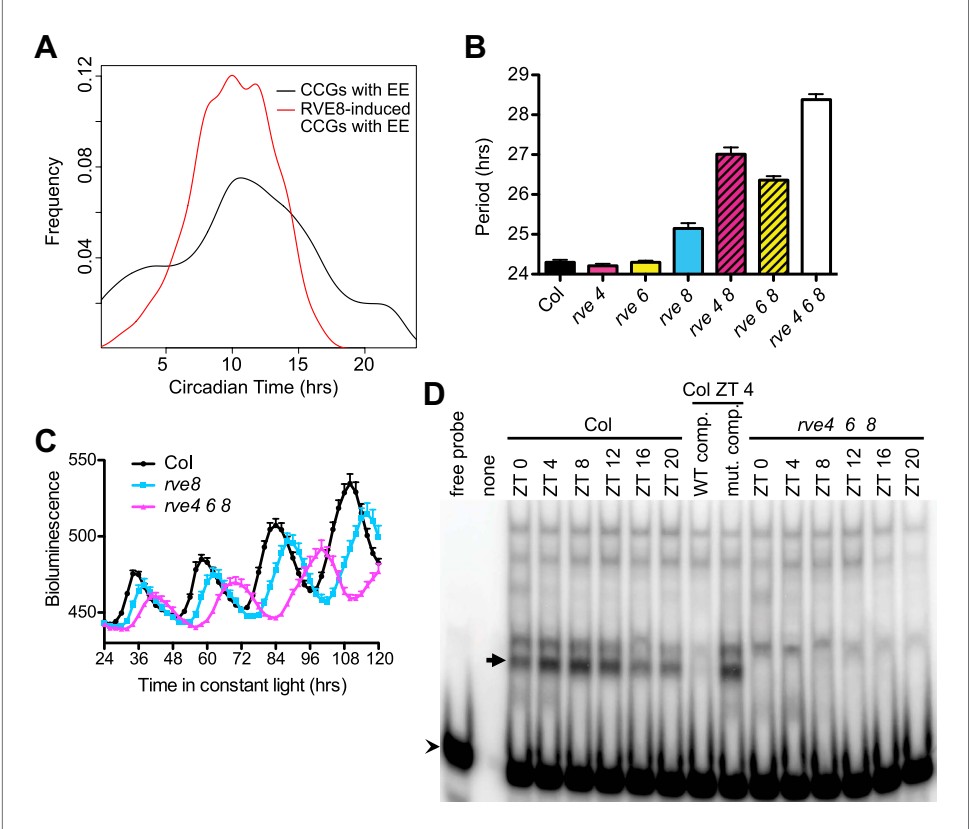

**Figure 4**. RVE8 functions through the EE. (**A**) Circadian phase distributions of all EE-containing CCGs and RVE8-induced EE-containing CCGs. The RVE8-induced EE-containing CCGs are enriched for an earlier phase than that of all EE-containing CCGs. The means of the phase distribution in these two groups (10.03 for RVE8-induced EE-containing CCGs; 10.75 for all EE-containing CCGs) are significantly different (p=0.007; Student's t-test). (**B**) Period of *CCR2::LUC* activity in *rve4-1*, *rve6-1* and *rve8-1* single, double and triple mutants. Seedlings were grown in LD for 6 days and released to constant red plus blue light. Mean ± SEM from 34 to 50 plants. (**C**) Circadian rhythms are lengthened but still robust in *rve4 rve6 rve8* mutants. Averaged bioluminescence of *CCR2::LUC* activity in Col, *rve8-1* and *rve4 rve6 rve8* triple mutants. Mean ± SEM from 20 to 25 plants. (**D**) An electrophoretic mobility shift (EMSA) assay with protein extracts made from Col and *rve4 rve6 rve8* plants grown in LD for 11 days. Plants were harvested at the indicated times. A 50-fold molar excess of unlabeled EE (WT competitor) or mutated EE (mutant competitor) double-stranded DNA was added as indicated. Arrow: the predominant afternoon EE-binding activity, arrowhead: unbound probe. See also *Figure 4—figure supplement 1*.

The following figure supplements are available for figure 4:

**Figure supplement 1**. Characterization of *RVE4*, *RVE6*, and *RVE8* mutant alleles.

Loss of RVE8 function has neither a strong effect on clock function nor on expression levels of evening-phased genes (*Farinas and Mas, 2011*; *Rawat et al., 2011*; *Hsu and Harmer, 2012*). This may be due to partial genetic redundancy, since there are four other close RVE8 homologs (RVE3, 4, 5, and 6) in the Arabidopsis genome (*Rawat et al., 2009*) and all of these proteins were found to bind to the EE (*Gong et al., 2008*; *Rawat et al., 2011*). To investigate whether RVE8 homologs play a partially redundant role with RVE8 in the circadian clock, we identified plants mutant for the two closest RVE8 homologs, *RVE4* and *RVE6* (*Figure 4—figure supplement 1A–C*) and examined clock function in single and higher order mutants. The pace of the clock in *rve4-1* and *rve6-1* single mutants is not significantly different from wild-type (*Figure 4B*). However, combining loss-of-function *RVE4* or *RVE6* alleles with *rve8-1* makes the period length much longer than *rve8-1* alone, while the *rve4 rve6 rve8* triple mutant has a period approximately 4 hr longer than wild-type (*Figure 4B*). These data suggest that *RVE4*, *6* and *8* play a partially redundant role in speeding up the pace of the clock. Despite the

severe long period phenotype, the *rve4 rve6 rve8* triple mutant displays robust circadian rhythms (*Figure 4C*).

We previously identified an afternoon-phased activity in plant extracts that specifically binds the EE, suggesting it might represent a cycling activator(s) for the EE (*Harmer and Kay, 2005*). Since we found RVE8 is an afternoon-phased activator of genes with EE in their promoters and that RVE4 and RVE6 play a partially redundant role with RVE8 in setting clock pace, we examined circadian-regulated EE-binding activity in the *rve4 rve6 rve8* triple mutant plants in an in vitro EE-binding assay. As expected, extracts from wild-type plants have an afternoon-phased EE binding activity (*Figure 4D*). Remarkably, this cycling EE-binding activity is abolished in the triple mutant (*Figure 4D*). Given that we have previously found that RVE4, RVE6 and RVE8 can all be affinity purified from plant extracts using EE sequences (*Rawat et al., 2011*), this strongly suggests that RVE4, 6 and 8 comprise a clock-regulated, evening-phased EE-binding activity.

## Transcripts of central clock genes are misregulated in *rve4 rve6 rve8* triple mutants

To further examine the functions of these RVEs (RVE4, 6 and 8) in plants, we examined the transcript profiles of genes we identified as RVE8 targets in the higher order *rve* mutants. In contrast to the *rve8-1* single mutant, which has normal expression levels of evening genes (*Rawat et al., 2011*; *Hsu and Harmer, 2012*), *rve6 rve8* double and *rve4 rve6 rve8* triple mutants grown in constant light (LL) display significantly reduced levels of *PRR5* transcripts (*Figure 5—figure supplement 1A*). Consistent with the progressively longer period in *rve6 rve8* and *rve4 rve6 rve8* mutants (*Figure 4B*), these mutants also have a greater delay in onset of *PRR5* transcript accumulation (*Figure 5—figure supplement 1A*).

We next examined expression levels of other clock genes in the triple mutant. The evening-phased genes, *PRR5* and *TOC1*, show a significant delay in onset of expression and reduced levels in the triple mutants compared to wild-type in both light/dark (LD) cycles (*Figure 5A,B*) and in constant light (LL) (*Figure 5E–F*). Although the peak phase of *PRR5* is not altered in the mutant in LD conditions (*Figure 5A*), the delay in the timing of increasing *PRR5* expression in *rve4 rve6 rve8* in the afternoon suggests that this is due to complex regulation of *PRR5* transcript levels by both light and the circadian clock. The morning-phased clock genes *CCA1* and *LHY* do not show any obvious differences in expression levels in *rve4 rve6 rve8* and wild-type plants during the day either when grown in LD (*Figure 5C,D*) or in LL (*Figure 5G,H*). However, these two morning-phased genes display slightly reduced transcript levels in the late night when grown in LD (ZT 21) (*Figure 5C,D*). This might be explained either by the long period phenotype of the *rve4 rve6 rve8* mutants or by elevated *TOC1* levels at the end of the night (*Figure 5B*) since TOC1 is a repressor of *CCA1* and *LHY* (*Gendron et al., 2012*; *Huang et al., 2012*; *Pokhilko et al., 2012*).

In addition to these expression level changes, de-synchronization between the evening and morning genes is observed in the *rve4 rve6 rve8* triple mutants. At the end of the second subjective day in LL (around ZT36), the peak times of *PRR5* and *TOC1* transcript accumulation are delayed approximately 6 hr in the triple mutant relative to wild-type (*Figure 5E,F*). In contrast, in these same samples, an approximately 3-hr phase delay is observed for trough levels of *CCA1* and *LHY* in the mutant relative to controls (*Figure 5K,L*). Similarly, at the third subjective morning (around ZT48), *PRR5* and *TOC1* trough levels display an approximately 9-hr phase delay in the triple mutant (*Figure 5I,J*) while an approximately 6-hr phase delay is observed in the peak expression levels for *CCA1* and *LHY* at that time (*Figure 5G,H*). This greater phase delay for evening compared to morning genes can be seen more clearly when plants grown in constant conditions are sampled at 1-hr intervals (*Figure 5M,O*). In addition, a significant change in the waveform of the evening gene *TOC1* (*Figure 5M*) but not the morning gene *LHY* (*Figure 5O*) is observed in the triple mutant in this high-resolution time course. Notably, the obvious change in the *TOC1* waveform is lost when these same data are plotted at 3-hr resolution (compare *Figure 5N,M*). Our data show that loss of multiple RVEs has an immediate effect on expression of evening genes and a delayed effect on morning genes, further supporting the main role of RVE8 as an activator of evening genes. Given the highly reticulated nature of the circadian network, altered expression of evening genes indirectly affects expression of morning genes.

Similarly reduced and delayed expression was also observed for *GI*, an evening-phased EE-containing clock gene, in LD and in LL (*Figure 5—figure supplement 1B,D*). Interestingly, the EE-containing day-phased gene, *PRR9* also showed reduced levels in LD (*Figure 5—figure supplement 1C*) and on the third day in LL (*Figure 5—figure supplement 1E*), consistent with *PRR9* being directly activated

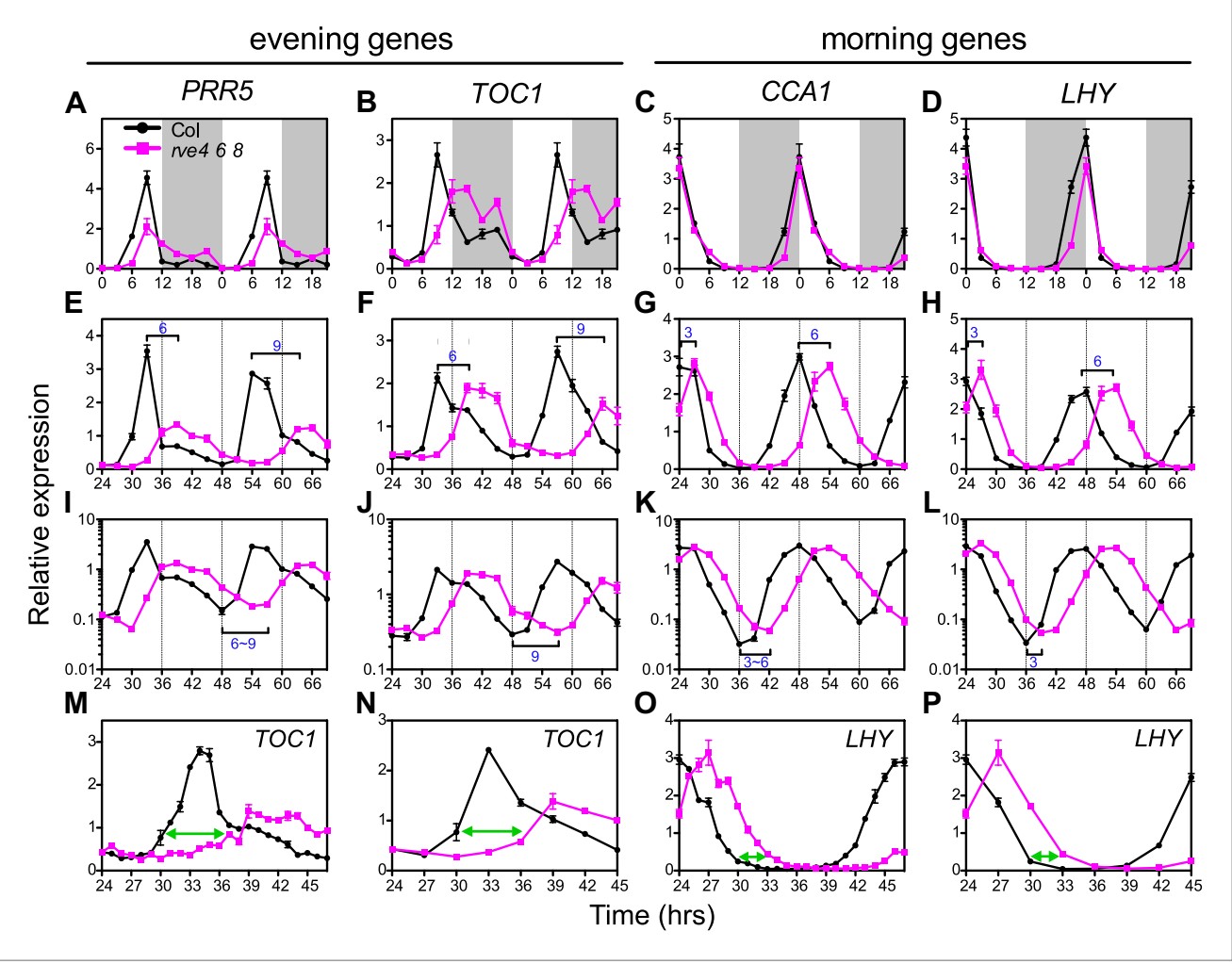

**Figure 5**. Expression of clock genes is altered in *rve4 rve6 rve8* triple mutants. (**A**), (**B**), (**E**), (**F**), (**I**), (**J**), (**M**) and (**N**) Expression of evening genes in Col and *rve4 rve6 rve8*. (**C**), (**D**), (**G**), (**H**), (**K**), (**L**), (**O**) and (**P**) Expression of morning genes in Col and *rve4 rve6 rve8*. (**A**)–(**D**) transcript levels in diurnal cycles. Seedlings were grown in LD for 7 days. White box: day, grey box: night. Data in (**A**–**D**) are double plotted to facilitate comparisons. (**E**)–(**P**) Transcript levels in LL. (**E**)–(**H**) Gene expression plotted on a linear scale. (**I**)–(**L**) The data shown in (**E**–**H**) are plotted with a log10 scale on the y-axis to better visualize differences in trough levels between the two genotypes. Horizontal brackets highlight the phase delay between Col and *rve4 rve6 rve8* mutants. (**M**)–(**P**) Transcript levels derived from a 1-hr resolution time course are presented with either every time point (**M** and **O**) or every third time point (**N** and **P**) displayed. Green arrows highlight the phase difference between Col and *rve4 rve6 rve8* mutants at ZT30. Transcript levels were determined by qRT-PCR and normalized to *PP2A*. Values represent mean ± SEM.

The following figure supplements are available for figure 5:

**Figure supplement 1**. Clock gene expression in wild type and the *rve4 rve6 rve8* mutant.

by RVE8 as suggested by the induction experiments carried out in the presence of an inhibitor of translation (***Figure 3—figure supplement 1E***).

## Expression of *RVE8* is controlled by other clock components, likely via PRR5, 7 and 9

To further explore regulatory interactions between RVE8 and other clock components, we examined *RVE8* expression in several clock mutants, including *toc1-4* (***Hazen et al., 2005a***), *lux-1* (***Hazen et al., 2005b***) and *CCA1-OX* (***Wang and Tobin, 1998***). *RVE8* expression is significantly reduced in all of these clock mutants in LD (***Figure 6A***), indicating that TOC1, LUX and CCA1 directly or indirectly regulate *RVE8* expression. Since TOC1, LUX, and CCA1 are thought to directly regulate expression of one or more of the *PRR5*, *7*, and *9* genes (***Farre et al., 2005***; ***Helfer et al., 2011***; ***Huang et al., 2012***), we

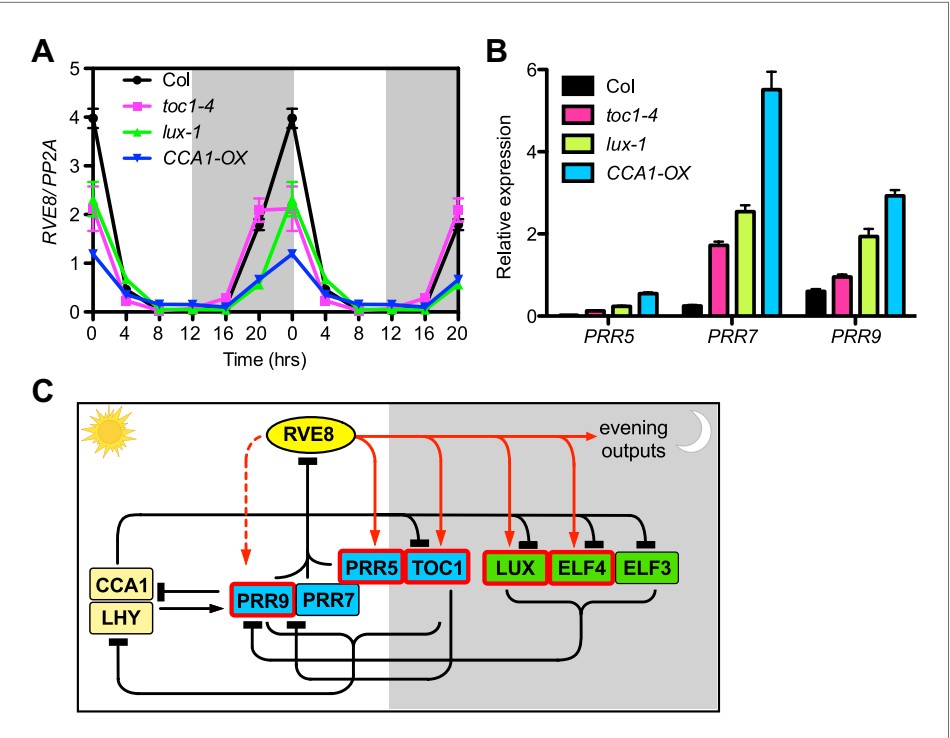

**Figure 6**. *RVE8* expression is likely controlled by other clock genes through *PRR5, 7* and *9*. (**A**) *RVE8* expression in Col, *toc1-4*, *lux-1* and *CCA1-OX* in LD. 7-day-old seedlings were collected at the times indicated and qRT-PCR was performed. Data are double-plotted to facilitate visualization. Values represent mean ± SEM. (**B**) Transcript levels of *PRR5*, *PRR7* and *PRR9* at ZT 0 (when *RVE8* transcript levels normally peak) in wild-type (Col), *toc1-4*, *lux-1*, and *CCA1-OX*. Expression levels are normalized to *PP2A*. Data are represented as mean ± SEM from three technical replicates. (**C**) A proposed clock model integrating RVE8 as an activator of evening clock genes. The relative time of action of each component during diurnal cycles is shown from left to right. White box: day, grey box: night. REVEILLE/CCA1/LHY family proteins are shown in yellow; pseudo-response regulators are shown in blue; the evening complex components are shown in green. Clock components with one or more EE in their promoter regions are marked with red boxes. Red solid arrow: activation, red dashed arrow: activation only displayed in specific condition (red arrows are based on the current study), black perpendicular bars: repression, black arrow: activation. In this study, we demonstrated that RVE8 directly activates multiple evening-phased clock and output genes and that *RVE8* is regulated by TOC1, LUX and CCA1, likely indirectly through their control of *PRR5, 7* and *9* expression. For clarity, only transcriptional regulation is represented.

hypothesized that reduced *RVE8* expression in the *toc1-4*, *lux-1* and *CCA1-OX* mutants might be due to up-regulation of the *PRRs*. Indeed, we found that at dawn (ZT0), when *RVE8* transcript levels normally peak (**Farinas and Mas, 2011**; **Rawat et al., 2011**), multiple *PRR* genes are up-regulated in each of these mutants (**Figure 6B**). These results are consistent with a model in which the PRRs directly repress *RVE8* expression and other clock genes indirectly control *RVE8* expression via regulation of *PRR5*, *7*, and *9* (**Figure 6C**). This model is supported by the increased *RVE8* expression seen in *prr5 prr7 prr9* mutants (**Rawat et al., 2011**) and the reported direct binding of PRR5 to the *RVE8* promoter (**Nakamichi et al., 2012**). Our findings that both *PRR5* and *PRR9* are directly activated by RVE8 (**Figure 3A**; **Figure 3—figure supplement 1E**) and that peak transcript levels of these genes are reduced in *rve4 rve6 rve8* mutants (**Figure 5A,E**; **Figure 5—figure supplement 1C,E**) further support the model that the PRRs and RVEs regulate each other to form a negative transcriptional feedback loop.

## Discussion

### RVE8 promotes expression of both central oscillator and output genes

Circadian rhythms coordinate numerous physiological and behavioral events with the appropriate time of day, in large part through genome-wide circadian regulation of gene expression (**Lowrey and**

*Takahashi, 2011*; *Farre and Weise, 2012*). Mechanisms governing the precise timing of circadian clock and output gene expression are therefore of great interest. It has previously been reported in both plants and mammals that central clock genes can directly regulate many output genes (*Gendron et al., 2012*; *Menet et al., 2012*; *Nakamichi et al., 2012*). Our finding that hundreds of clock-regulated genes are induced or repressed upon induction of RVE8 suggests RVE8 is an important regulator of both the clock itself and output pathways.

It seems likely that RVE8 is a direct activator of many evening-phased genes. EE sequences are significantly enriched among RVE8-induced targets relative to all evening-phased genes (*Table 1A*). In addition, RVE8 binds to EE sequences in vivo and in vitro (*Gong et al., 2008*; *Rawat et al., 2011*) and plants mutant for *RVE8* and its close homologs *RVE4* and *RVE6* have lost an afternoon-phased EE binding activity (*Figure 4D*). Furthermore, for all genes tested, activation of evening-phased genes by RVE8 does not require new protein synthesis (*Figure 3*). In contrast, genes repressed upon induction of RVE8 activity are primarily morning-phased and are not enriched for any promoter motif relative to all clock-regulated morning-phased genes (*Table 1B*), suggesting RVE8 regulates these genes indirectly. In support of a largely indirect role for RVE8 in repression of gene expression, inhibition of translation reduced or eliminated decreases in gene expression upon RVE8 induction (*Figure 3—figure supplement 1D–G*). Thus RVE8 is unique among Arabidopsis clock genes in that it acts primarily, and perhaps even exclusively, as an activator of gene expression.

Unlike CCA1 and LHY, which were shown to have similar activity at different time of day in ethanol-inducible lines (*Knowles et al., 2008*), we have found that RVE8 activity is gated with maximum activity in the afternoon (*Figure 1D*). This discrepancy in gating regulation may explain why over-expression of *CCA1* or *LHY* causes arrhythmicity (*Schaffer et al., 1998*; *Wang and Tobin, 1998*) while overexpression of *RVE8* instead causes an advanced phase and short period phenotype (*Rawat et al., 2011*).

## RVE8 shapes evening phase

*RVE8* transcript levels peak at dawn, but RVE8 protein levels peak in the subjective afternoon (*Rawat et al., 2011*). Most RVE8-induced transcripts have a peak circadian phase between CT8 and CT12 (*Figure 2G*), approximately 2–6 hr after the peak phase of RVE8 protein levels. This delay in RVE8 target gene transcript accumulation relative to RVE8 protein might be explained by antagonistic regulation of target genes by RVE8 and the cycling repressors CCA1 and LHY. CCA1 and LHY protein levels peak in the subjective morning (*Wang and Tobin, 1998*; *Kim et al., 2003*), well before RVE8. A mathematical model investigating the consequences of oppositely acting transcription factors on regulation of a common target gene predicted that when the phase of a cycling transcriptional repressor precedes that of a cycling transcriptional activator ('repressor-precedes activator'), the peak phase of expression of the output would occur after that of the activator (*Ueda et al., 2005*).

The genes both induced by RVE8 and containing EE motifs in their promoter regions are the most likely direct targets of RVE8. Although most clock-controlled genes containing an EE have an evening phase, the EE-containing RVE8-induced genes are more specifically enriched for an early-evening phase (*Figure 4A*), suggesting that RVE8 controls a subset of EE-containing genes. How these RVE8 targets are distinct from the rest of the EE-containing CCGs remains unclear. The clock may fine-tune expression of EE-containing genes through the action of multiple clock-controlled promoter motifs, generating the wide range of phases seen across all EE-containing genes (*Figure 4A*). For example, it has been reported that a combination of morning-, day-, and night-phased DNA elements generates the day-phased expression of *Cry1* in mammalian cells. In this case, the strength of night-phased repressors relative to the day-phased activators modulates the extent of phase delay (*Ukai-Tadenuma et al., 2011*).

The RVE8 homologs RVE4, 5 and 6 have also been found associated with the EE in extracts made from plants harvested in the afternoon, suggesting that they might act in a similar manner to RVE8 (*Rawat et al., 2011*). This possibility is supported by the further lengthening in circadian period seen in higher order mutants combining *rve4* or *rve6* with *rve8* (*Figure 4B*), suggesting these factors play partially redundant roles in speeding up the pace of the clock. The loss of afternoon-phased EE binding activity seen in the *rve4 rve6 rve8* triple mutants but not in *rve8* single mutants (data not shown) suggests these RVEs contribute to the activity of the clock-regulated afternoon-phased EE activator.

## The long period in *rve4 rve6 rve8* is likely due to delayed expression of evening genes

Among the evening-phased central clock genes examined, all show significantly reduced and delayed expression in LD and in LL in *rve4 rve6 rve8* (*Figure 5* and *Figure 5—figure supplement 1*). The long period in *rve4 rve6 rve8* mutants might in principle be due either to a decrease in peak levels or a delay in onset of expression of evening genes. However, consideration of the phenotypes of plants mutant for various evening-phased clock genes makes us favor the latter possibility. *toc1* and *prr5* mutants have short-period phenotypes (*Strayer et al., 2000*; *Eriksson et al., 2003*; *Yamamoto et al., 2003*); loss of *GI* causes a short period in most conditions (*Park et al., 1999*; *Mizoguchi et al., 2005*; *Martin-Tryon et al., 2007*); and *lux* and *elf4* mutants are arrhythmic (*Doyle et al., 2002*; *Onai and Ishiura, 2005*; *Hazen et al., 2005b*). Therefore, reduced expression of any of these EE-containing evening genes is unlikely to cause the long period phenotype displayed by *rve4 rve6 rve8*. On the other hand, the delayed phase of expression of clock genes can first be observed in evening-phased genes and only later in morning-phased genes (*Figure 5*). This suggests that the long period seen in *rve4 rve6 rve8* is mainly caused by delayed expression of evening genes, which then indirectly causes a delayed phase of expression of morning genes. In support of this idea, in *RVE8*-overexpressing plants (which have a short-period phenotype), the peak phase of expression of *TOC1* is clearly advanced soon after plants are released into free-run whereas phase advances are not seen for the morning-phased genes *CCA1* and *LHY* until much later (*Rawat et al., 2011*). Similarly, delays in the phase of post-transcriptional processes have previously been suggested to contribute to long-period phenotypes in animals (*Rothenfluh et al., 2000*; *Syed et al., 2011*).

## The EE is a regulatory nexus crucial for clock function

Most clock components in Arabidopsis are either regulated by the EE (including most evening-phased genes and one day-phased gene, *PRR9*) or regulate the EE (two morning-phased components, CCA1 and LHY, and the afternoon-phased activator, RVE8) (*Figure 6C*). However, plants mutant for *CCA1* and *LHY*, the sole previously defined circadian regulators of EE-containing clock genes, have persistent circadian rhythms, albeit with a short period (*Alabadi et al., 2002*; *Mizoguchi et al., 2002*; *Locke et al., 2005*). Our discovery that RVE8 and its homologs are activators of the EE may explain the rhythmicity of *cca1 lhy* mutants. As modeled using Ueda et al's 'repressor-precedes-activator' formula (*Ueda et al., 2005*), inhibition in the morning by CCA1 and LHY and activation by RVE8 in the afternoon would result in rhythmic expression of EE target genes with peak expression delayed relative to peak RVE8 protein levels. Reduction or loss of activity of the cycling repressor function (*CCA1/LHY*) would result in a phase advance, causing earlier expression of EE-containing target genes, but rhythms would persist due to clock-regulated RVE8 activity. Such a phase advance and consequent short-period phenotype is indeed observed in *cca1* and *lhy* single and double mutants (*Green and Tobin, 1999*; *Mizoguchi et al., 2002*).

Interestingly, CCA1/LHY and RVE8 contain a similar Myb-like DNA binding domain and belong to the same family of transcription factors (*Rawat et al., 2009*, *2011*). Even though they have distinct biochemical functions, with CCA1 and LHY serving as repressors and RVE8 as an activator of EE-containing genes, both CCA1/LHY and RVE8 are responsible for shaping the circadian pattern of expression of evening-phased genes. This joint regulation of common targets may explain why circadian rhythms persist upon mutation of the repressor Mybs or the activator Mybs alone.

## The plant circadian clock consists of a highly interconnected, complex network

Current models of the plant clock suggest that it is composed of transcription factors that are primarily repressors of gene expression which interact to form interlocked morning and evening feedback loops (*Gendron et al., 2012*; *Huang et al., 2012*; *Pokhilko et al., 2012*). However, our findings substantially revise this view. We have demonstrated that the RVEs are an integral part of the circadian oscillator but are primarily (and perhaps exclusively) activators of gene expression. In addition, our findings suggest that the view of the plant clock as constituted of coupled morning and evening transcriptional feedback loops is inadequate. RVE8 itself, with its morning-phased peak in transcript levels but afternoon-phased peak in protein levels (*Rawat et al., 2011*), doesn't fit neatly into either the 'morning' or 'evening' category. Furthermore, the highly interconnected nature of the regulatory interactions underlying the plant clock (*Figure 6C*) make it virtually impossible to identify discrete regulatory feedback

loops and suggest that the plant clock is best viewed as a highly interconnected, complex regulatory network.

## Materials and methods

### DNA and plant materials

The *RVE8::RVE8:GR* construct was created using a PCR fusion-based approach (*Hobert, 2002*), placing a 2.5 kb genomic fragment of *RVE8* (containing ~0.7 kb upstream of the translational start site) and a 1.7 kb DNA fragment containing the GR coding sequence and OCS 3′ from pART7-GR (donated by John Harada) together. The PCR fusion product was then cloned into the NotI site in the binary vector pML-BART. The *35S::RVE8:GR* construct was created by cloning *RVE8* coding sequence into pART7-GR via XhoI and SmaI sites, and then subcloning into the NotI site in the binary vector pML-BART. The *RVE8::RVE8:GR* and *35S::RVE8:GR* clones were transformed into *rve8-1 CCR2::LUC+* via the floral dip method (*Zhang et al., 2006*). Homozygous single-insertion site transformants were selected based on BASTA resistance in the T2 and T3 generations.

T-DNA insertion mutants *rve4-1* (Salk_137617) and *rve6-1* (Salk_069978) (*Alonso et al., 2003*) were obtained from the Arabidopsis Biological Resources Center. Homozygous mutants were identified by PCR of genomic DNA using primers flanking the insertion site and complementary to the T-DNA left border (primers are listed in *Supplementary file 3*). *rve4-1* and *rve6-1* were crossed to *rve8-1 CCR2::LUC+* to generate *rve4 rve8* and *rve6 rve8* double mutants and *rve4* and *rve6* single mutants, all carrying the *CCR2::LUC+* reporter. The *rve4 6 8* triple mutant was created by crossing *rve4 rve8 CCR2::LUC+* and *rve6 rve8 CCR2::LUC+*.

*lux-1*, *toc1-4* and *CCA1-OX* were previously described (*Wang and Tobin, 1998*; *Hazen et al., 2005a*, *2005b*).

### Dexamethasone (DEX) and/or cycloheximide (CHX) treatment

*rve8-1* and *rve8-1 RVE8::RVE8:GR* seeds were sterilized and stratified on fine nylon mesh (Small Parts, Logansport, IN; 100 µM 44%) on Murashige and Skoog (MS) agar media containing 3% sucrose at 4°C in the dark for 2 days. The seedlings were grown under 12-hr light:12-hr dark condition with 50–60 µmol/m²/s white fluorescent light at 22°C for 7–8 days. At ZT4 (4 hr after lights on), the mesh and seedlings were transferred to liquid MS media containing 3% sucrose with 30 µM DEX (Sigma D1881, St. Louis, MO; 60 mM DEX stock solution was made in ethanol and stored at −20°C) or 0.05% ethanol (mock treatment). For cycloheximide treatment, 200 µM CHX (Sigma C4859; stock solutions were 100 µg/µl in DMSO) or 0.056% DMSO (mock treatment) was added at the time of DEX or ethanol mock treatment. After 2 or 4 hr incubation as indicated with gentle agitation, plants were quickly harvested, frozen in liquid nitrogen and stored in −80°C until processed.

### RNA isolation and RNA-seq library construction

Total RNA from three biological replicates (~30 plants each) for each condition was isolated using Trizol (Invitrogen, Grand Island, NY), treated with DNase (Qiagen, Germantown, MD), and purified using the RNeasy MinElute Cleanup Kit (Qiagen). The quality of the isolated total RNA was determined by NanoDrop ND 1000 (NanoDrop Technologies, Wilmington, DE). Samples with both a 260 nm:280 nm ratio and a 260 nm:230 nm ratio between 2 and 2.3 were processed further. The RNA-seq libraries were prepared using a customized Illumina-based strand-specific multiplex library construction protocol modified from *Wang et al. (2011)*. Briefly, mRNA was isolated from 8 µg of total RNA using Dynabeads mRNA DIRECT Kit (Invitrogen) and fragmented to ~200 nucleotide pieces. After the first strand cDNA synthesis was carried out using random primers, the second strand cDNA was synthesized using a special dNTP mix in which dTTP is replaced by dUTP. Following end-repair (Y9140-LC-L; Enzymatics, Beverly, MA) and addition of a dA to the 3′ end, both ends of cDNA were ligated with Y-shaped adaptors containing an index unique to each library. The second strand cDNA was then digested using Uracil DNA glycosylase (Enzymatics). Primers partially complementary to the adaptor sequences were used to amplify the libraries for 12 PCR cycles using High-Fidelity Polymerase (Phusion, Ipswich, MA). The libraries were further size-selected using a 1:1 volume of AMPure XP beads (Beckman Coulter, Brea CA). The size and quality of resulting libraries were examined using a Bioanalyzer 2100 (Agilent, Santa Clara, CA). The 12 libraries were then quantified by qPCR and equally pooled for 2 lanes of single end 50 bp sequencing in HiSeq 2000 machine (Illumina, San Diego, CA). The adaptors containing index sequences and primers used for amplification are listed in *Supplementary file 3*.

## Quality filtering and alignment of RNA-seq data

The raw reads (~310.3 million reads) were initially subjected to quality filtering to remove low quality reads using the FASTX-toolkit (*Pearson et al., 1997*) with the following parameters (−q 20, minimum quality score to keep: 20; −p 85, minimum percent of bases that must satisfy the quality score cut-off: 85). A custom perl script was then used to remove Illumina adapter sequences from the resulting reads. The reads were then separated by their custom barcode sequences (de-multiplexing) using Fastx_barcode_splitter (included in the FASTX toolkit) allowing up to one mismatch per barcode. 16 to 22 million reads per libraries were obtained and aligned against the Arabidopsis cDNA representative_gene_model (TAIR 10) using BWA (*Li and Durbin, 2009*) and Samtools (*Li et al., 2009*). (The parameter used to map the reads for BWA was aln -l 20.) The resulting BAM files from the two lanes were merged using Samtools and then converted to SAM files. The reads from these SAM files were then separated based on their alignment to the forward or reverse strand. Only the reads mapped to the reverse strand were used to calculate the read counts using a custom R script, and these counts were then used in analysis of differential expression.

## Differential expression analysis of RNA-seq data

edgeR was used to generate the pseudo-normalized counts for visualization and to carry out differential gene expression analysis (*Robinson et al., 2010*) using R 2.14.1 (*R Development Core Team, 2011*). Transcripts that have at least one count per million in at least three samples were considered expressed genes and kept for downstream analysis. Exact tests were performed using tagwise dispersion and the prior n was set to 6.25. FDR 0.01 was used as a cut-off for differentially expressed genes. Genes significantly differentially expressed between the mock- and DEX-treated transgenic line (*rve8 RVE8::RVE8:GR*), or between the DEX-treated *rve8 RVE8::RVE8:GR* and *rve8* plants, were grouped into RVE8-induced or RVE8-repressed genes. Genes that are responsive to DEX treatment in *rve8* mutant (i.e., the genes significantly differentially expressed between '*rve8* + DEX' and '*rve8* + mock') were removed from the RVE8-induced and RVE8-repressed gene lists. Only the genes uniquely induced or repressed by RVE8 (i.e., not showing the same trend in *rve8*) were used for further analysis. The significant gene sets (both RVE-regulated or DEX-regulated) are listed in *Supplementary file 1C–F*.

## Phase and motif analysis for RVE8 target genes

Circadian phases of the 583 RVE8-induced genes and 850 RVE8-repressed targets were determined in a previous study using JTK_CYCLE (*Hsu and Harmer, 2012*). The 376 RVE8-induced cycling genes (64% of the induced genes) and 525 RVE8-repressed cycling genes (62% of the repressed genes) were subjected to phase and motif analysis. Distributions of the phases of the RVE8-induced and -repressed clock-regulated genes were plotted using the density function in R (*R Development Core Team, 2011*). Overrepresented motifs in the promoters of RVE8-regulated genes were identified using the SCOPE motif finder (*Carlson et al., 2007*). Fixed regions of 1500 bp upstream of the translational start site (corresponding to both strands) of RVE8-regulated genes were used for computation of significance compared to all the genes in the genome. Significance is the negative logarithm of expectation. Significance greater than zero is statistically meaningful; the larger the significance value, the higher its statistical significance. Coverage indicates the percentage of genes that have at least one occurrence of the motif in question. The fractions of genes containing the top-scoring motifs among the evening-phased (CT 8 to CT 14) and morning-phased (CT 20 to CT 2) RVE8 targets were compared to the fractions found in all of the clock-regulated genes in the corresponding phase group. Fisher's exact test was performed in R to examine if the presence of these motifs in RVE8 targets is enriched compared to their presence in the evening- and morning-phased genes.

## Analysis of gene expression (qRT-PCR)

For gene expression in diurnal cycles, around 30 seedlings per sample were grown under 12 hr white light (50–60 μmol/m²/s, generated using cool white fluorescent bulbs):12 hr dark at 22°C for 7 days and harvested at the times indicated. For gene expression in free-run, seedlings were released to constant white light after entrainment in diurnal cycles for 7 days, and harvested at the times indicated. RNA was isolated using Trizol (Invitrogen) and was then treated with DNase (Qiagen). cDNA was synthesized using SuperScriptase II (Invitrogen) following the manufacturer's protocol. qRT-PCR was performed as previously described (*Martin-Tryon et al., 2007*). Three technical triplicates for each sample were run using iQ5 Real Time PCR machine (Bio-Rad, Hercules, CA), and starting quantity was estimated from

critical thresholds using the standard curve method. Data for each sample were normalized to the respective *PROTEIN PHOSPHATASE 2A* (*PP2A*) expression level. The primer sets for each transcript are listed in ***Supplementary file 3***.

## Luciferase imaging

Luciferase imaging was performed and analyzed as previously described (***Martin-Tryon et al., 2007***). Seedlings were entrained in 12 hr white light (50–60 µmol/m²·/s; cool white fluorescent bulbs):12 hr dark at 22°C for 6 days before being released to constant red plus blue light (33µEi red light, 20µEi blue light) for luciferase activity analysis using an ORCA II ER (Hamamatsu, Bridgewater, NJ) CCD camera. Illumination was provided by monochromatic red and blue LED lights (XtremeLux, Santa Clara, CA). Images were analyzed using MetaMorph (Molecular Devices, Sunnyvale, CA) and free-running periods were estimated using Fast Fourier Transform Non-Linear Least Squares (***Plautz et al., 1997***).

## Electrophoretic mobility shift assay (EMSA)

11-day-old seedlings grown in 12 hr white light (50–60 µmol/m²/s; cool white fluorescent bulbs):12 hr dark cycle at 22°C were harvested at the times indicated. Plant whole-cell extracts were made and the electrophoretic mobility shift assay was performed as previously described (***Harmer and Kay, 2005***). Briefly around 1.5 g of tissue per sample was harvested, frozen in liquid nitrogen immediately and stored at −80°C until processed. The frozen tissue was ground to a fine powder, suspended in homogenization buffer (15 mM HEPES, pH 7.6, 40 mM KCl, 5 mM MgCl$_2$, 1 mM DTT, 0.1 mM PMSF, and 1X complete protease inhibitor cocktail) and $(NH_4)_2SO_4$ was added to 0.4 M. The insoluble components were pelleted by ultracentrifugation and removed, then solid $(NH_4)_2SO_4$ was added to the supernatant to ~90% saturation. Proteins were pelleted by ultracentrifugation, resuspended in resuspension buffer (20 mM HEPES, pH 7.6, 40 mM KCl, 0.1 mM EDTA, 10% glycerol, 1 mM DTT, 0.1 mM PMSF, and 1× complete protease inhibitor cocktail), and dialyzed using dialysis cartridges (7000 MWCO Slide-A-Lyzer; Pierce, Rockford, IL) against dialysis buffer (20 mM HEPES, pH 7.2, 40 mM KCl, 0.1 mM EDTA, 10% glycerol, 2.5 mM DTT, 0.1 mM PMSF). The dialyzed proteins were quantified, aliquoted and saved at −80°C until used. 15 µg of the dialyzed protein was incubated with 20 fmol of radiolabelled double-stranded DNA containing the EE and flanking sequences from the *CCR2* promoter in reaction buffer (20 mM HEPES, pH 7.2, 80 mM KCl, 0.1 mM EDTA, 10% glycerol, 2.5 mM DTT, 8 ng/µl poly [dI-dC]) with or without the competitors as indicated for 15 min at room temperature. A 50-fold molar excess of unlabeled *CCR2*-EE (WT competitor) or mutated *CCR2*-EE (mutated competitor) DNA was added as indicated for binding-specificity control. The binding assays were resolved by electrophoresis on 5% non-denaturing polyacrylamide gels. The dried gel was imaged using a Storm PhosphorImager (Molecular Dynamics, Sunnyvale, CA). The probe and competitor DNA sequences are listed in ***Supplementary file 3***.

## Acknowledgements

We thank J Maloof and M Covington for providing scripts for RNA-seq analysis; J Harada for donating the GR vector; R Kumar for providing the indexed adapters; S Brady and V Sundaresan for thoughtful comments on the project; M Covington and K Nozue for helpful advices on data analysis; H Wu for critical reading of the manuscript and numerous helpful suggestions on the project; and all the Harmer and Maloof laboratory members for valuable discussions. Sequencing data have been deposited in the Gene Expression Omnibus database (GEO) (http://www.ncbi.nlm.nih.gov/geo) with accession number GSE38879.

## Additional information

### Funding

| Funder | Grant reference number | Author |
| --- | --- | --- |
| National Institutes of Health | GM069418 | Stacey Harmer |
| Taiwan Merit Scholarship | NSC-095-SAF-I-564-014-TMS | Polly Yingshan Hsu |

The funders had no role in study design, data collection and interpretation, or the decision to submit the work for publication.

## Author contributions
PYH, SLH, Conception and design, Acquisition of data, Analysis and interpretation of data, Drafting or revising the article; UKD, helped optimize the strand-specific library construction protocol and assisted with the low-level RNA-seq analysis

# Additional files

### Supplementary files
• Supplementary file 1. RNA-seq analysis of RVE8-regulated genes. (**A**) Pipeline and summary of RNA-seq data analysis. (**B**) Pseudo-normalized counts for visualization. Counts for each library were normalized (*Robinson et al., 2010*) according to the library size and represented as counts per million. (**C**) Genes identified as significantly induced by RVE8 by RNA-seq. Genes significantly differentially expressed (adjusted p<0.01) between *RVE8:GR* + DEX and *RVE8:GR* + mock or between *RVE8:GR* + DEX and *rve8* + DEX, but not between *rve8* + DEX and *rve8* + mock, are defined as RVE8-induced or RVE8-repressed targets. (**D**) Genes identified as significantly repressed by RVE8 by RNA-seq. Genes significantly differentially expressed (adjusted p<0.01) between *RVE8:GR* + DEX and *RVE8:GR* + mock or between *RVE8:GR* + DEX and *rve8* + DEX, but not between *rve8* + DEX and *rve8* + mock, are defined as RVE8-induced or RVE8-repressed targets. (**E**) Genes identified as significantly induced by DEX by RNA-seq. Genes significantly differentially expressed between *rve8* + DEX and *rve8* + mock are defined as DEX-induced or DEX-repressed genes (adjusted p<0.01). (**F**) Genes identified as significantly repressed by DEX by RNA-seq. Genes significantly differentially expressed between *rve8* + DEX and *rve8* + mock are defined as DEX-induced or DEX-repressed genes (adjusted p<0.01). (**G**) Clock genes identified as RVE8 targets in the RNA-seq experiment. Genes found to be differentially expressed in response to RVE8 induction (adjusted p<0.01). Evening-phased genes are highlighted in yellow. (**H**) Significantly overrepresented functional classifications among RVE8-induced genes. The enrichment of the functional terms was identified using BioMaps in Virtual Plant 1.2 (http://virtualplant.bio.nyu.edu/cgi-bin/vpweb/) (*Katari et al., 2010*). Functional classifications were provided by the Munich Information Center for Protein Sequences (MIPS) (*Schoof et al., 2005*). All genes classified as expressed in the RNA-seq experiment were used for the background. Fisher's exact test (with FDR correction) was performed and the cut-off value for statistical significance was set to 0.01.

• Supplementary file 2. Promoter motifs overrepresented in RVE8-regulated CCGs (related to *Table 1*). Promoters were defined as the 1500 bp region upstream of the translational start site and motifs were identified using the *SCOPE* motif finder (*Carlson et al., 2007*). Both strands were considered for calculation of significance. Background frequency was determined using all genes in the genome. (**A**) Up-regulated CCGs (376 genes). (**B**) Down-regulated CCGs (525 genes).

• Supplementary file 3. Primers used in this study.

### Major datasets
The following dataset was generated:

| Author(s) | Year | Dataset title | Dataset ID and/or URL | Database, license, and accessibility information |
|---|---|---|---|---|
| Hsu Polly Yingshan, Harmer Stacey L | 2012 | Identification of RVE8 target genes | GSE38879, http://www.ncbi.nlm.nih.gov/geo/query/acc.cgi?acc=GSE38879 | Publicly available at the Gene Expression Omnibus database (GEO) (http://www.ncbi.nlm.nih.gov/geo). |

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
