## [Decision Letter]

Thank you for choosing to send your work entitled “Accurate timekeeping is controlled by a cycling activator in Arabidopsis” for consideration at *eLife*. Your article has been evaluated by a Senior editor, Detlef Weigel, and 2 reviewers, one of whom, Joanne Chory, is a member of our Board of Reviewing Editors. The reviewers were enthusiastic about your studies and found them to be very timely. We are thus is pleased to provisionally accept your paper for publication in *eLife,* subject to you performing one additional experiment.

After consultation, the reviewers agreed that the major unresolved issue is that the triple *rve* mutant is not arrhythmic, precluding you from concluding whether these proteins are integral members of the clock, or more accurately, should be considered an input of clock activity. The final test would require having a sextuple mutant (*cca1 lhy rve8 rve6 rve5 rve4*) and then there would be two predictions. 1) The mutant is arrhythmic, making a convincing case that all the members of this set of single Myb domain proteins are key clock components, essential for generating circadian rhythms. 2) Alternatively, the shortening and lengthening period effects of the CCA1 and RVE8 subsets are compensated and you might conclude that these Mybs simply modulate the clock, presumably linking it to the light input. The reviewers felt it would be extremely important to be able to test the above possibilities with the sextuple knock out, but this is perhaps outside the range of the present manuscript. Rather, the following solution is offered: to strengthen your case for desynchronization of morning/evening genes, the reviewers are asking you to sample additional time points of the triple KO vs wt for a single representative each of the morning and evening genes (e.g., CCA1 and TOC1).

A second, minor point was mentioned by one of the reviewers. To quote:

“The current view in the field is that CCA1 and LHY are both repressors of evening genes such as PRR1, and activators of morning genes such as PRR9 and PRR7. So, the current model still has one set of activators, which in addition are positively regulated by light. The authors indicate in the text that the actual model is ‘mostly’ based on repressors, so I understand that you are acknowledging that at least one set of activators are present. The information added is substantial anyway. So far, only the morning loop had activators (CCA1 and LHY activating PRR9 and PRR7), the evening loop had only repressors, and both loops were connected through repressors. The results of the genomic analysis of RVE8 targets, combined with knowledge that RVE8 binds to the EE, and the bioinformatic evaluation of the distribution of the EE among targets, shows that RVE8 is mainly a direct activator of evening genes and an indirect repressor of morning genes. This makes RVE8 and its close homologs RVE4 and RVE6 the only components of the clock that positively connect morning and evening loops. The authors should state the novelty of their finding a bit more explicitly, so they avoid leading the readers to the erroneous conclusion that the current clock is ‘exclusively’ based on repressors.”

---

## [Author Response]

*After consultation, the reviewers agreed that the major unresolved issue is that the triple rve mutant is not arrhythmic, precluding you from concluding whether these proteins are integral members of the clock, or more accurately, should be considered an input of clock activity. The final test would require having a sextuple mutant* (cca1 lhy rve8 rve6 rve5 rve4) *and then there would be two predictions. 1) The mutant is arrhythmic, making a convincing case that all the members of this set of single Myb domain proteins are key clock components, essential for generating circadian rhythms. 2) Alternatively, the shortening and lengthening period effects of the CCA1 and RVE8 subsets are compensated and you might conclude that these Mybs simply modulate the clock, presumably linking it to the light input. The reviewers felt it would be extremely important to be able to test the above possibilities with the sextuple knock out, but this is perhaps outside the range of the present manuscript. Rather, the following solution is offered: to strengthen your case for desynchronization of morning/evening genes, the reviewers are asking you to sample additional time points of the triple KO vs wt for a single representative each of the morning and evening genes (e.g., CCA1 and TOC1)*.

We thank the reviewers for the suggestion of creating the sextuple mutant for future experiments. This is certainly one of our long-term goals.

We also thank the reviewers for suggesting sampling additional time points. We sampled a circadian time course with one-hour resolution, which greatly improved the visualization of de-synchronization between the morning and evening genes, both with regards to timing of peak expression and alterations of waveform in the *rve* triple mutant (Figure 5). We feel this substantiates our conclusion that evening clock genes are direct targets of RVE activity, whereas morning genes are indirect targets.

*“The current view in the field is that CCA1 and LHY are both repressors of evening genes such as PRR1, and activators of morning genes such as PRR9 and PRR7. So, the current model still has one set of activators, which in addition are positively regulated by light. The authors indicate in the text that the actual model is ‘mostly’ based on repressors, so I understand that you are acknowledging that at least one set of activators are present. The information added is substantial anyway. So far, only the morning loop had activators (CCA1 and LHY activating PRR9 and PRR7), the evening loop had only repressors, and both loops were connected through repressors. The results of the genomic analysis of RVE8 targets, combined with knowledge that RVE8 binds to the EE, and the bioinformatic evaluation of the distribution of the EE among targets, shows that RVE8 is mainly a direct activator of evening genes and an indirect repressor of morning genes. This makes RVE8 and its close homologs RVE4 and RVE6 the only components of the clock that positively connect morning and evening loops. The authors should state the novelty of their finding a bit more explicitly, so they avoid leading the readers to the erroneous conclusion that the current clock is ‘exclusively’ based on repressors.*”

We agree with the reviewer and we have revised the manuscript to highlight how our findings fundamentally alter our understanding of the plant clock. We have revised the Abstract and included a new paragraph in the Discussion (“The plant circadian clock consists of a highly interconnected, complex network”) to emphasize our findings more explicitly. Specifically, we demonstrated that these RVEs are primarily (and perhaps exclusively) activators of gene expression. Further, our findings suggest that it is better to view the plant clock as a highly interconnected complex network rather than consisting of coupled morning and evening transcriptional feedback loops.